# Correlation between Antibiotic Consumption and Resistance of Invasive *Streptococcus pneumoniae*

**DOI:** 10.3390/antibiotics10070758

**Published:** 2021-06-22

**Authors:** Milan Čižman, Verica Mioč, Tom Bajec, Metka Paragi, Tamara Kastrin, José Gonçalves

**Affiliations:** 1Department of Infectious Diseases, University Medical Centre, 1000 Ljubljana, Slovenia; 2Department for Public Health Microbiology, National Laboratory of Health, Environment and Food, 1000 Ljubljana, Slovenia; verica.mioc@nlzoh.si (V.M.); metpar1@nlzoh.si (M.P.); tamara.kastrin@nlzoh.si (T.K.); joscar1@nlzoh.si (J.G.); 3Tomtim d.o.o, 1000 Ljubljana, Slovenia; tbajec@siol.net; 4Institute of Sustainable Processes, University of Valladolid, 47011 Valladolid, Spain

**Keywords:** outpatients, antibiotic, consumption, resistance, *S. pneumoniae*, correlation, Slovenia, prescription rate

## Abstract

There is a lack of long-term studies that correlate different metrics of antibiotic consumption and resistance of invasive *S. pneumoniae.* The present study aims to investigate the correlation between national outpatients total antibiotic, penicillin and broad spectrum penicillins consumption expressed in daily doses per 1000 inhabitants per day (DID) with the ATC/DDDs, WHO version of 2019 (new version) and 2018 (old version), number of prescriptions per 1000 inhabitants per year (RxIDs) and number of packages per 1000 inhabitant per day (PIDs) with the resistance of invasive *S. pneumoniae* in Slovenia in the period from 2000 to 2018. The prevalence of penicillin resistance of invasive *S. pneumoniae* decreased by 47.13%, from 19.1% to 10.1%. Decline of resistance showed the highest correlation (R = 0.86) between RxIDs followed by PID (R = 0.85) and resistance of *S. pneumoniae*. Higher correlation between total use of antibiotics expressed in DID WHO version 2019 (R = 0.80) than for WHO version 2018 (R = 0.78) was found. Very high (R = 0.84) correlation between use of β-lactams expressed in PID, and RxIDs (R = 0.82) and reasonable (R = 0.59) correlation expressed in DIDs version 2019 was shown as well. The consumption of broad -spectrum penicillins (J01CA and J01CR02) expressed in PID (R = 0.72) and RxIDs (0.57) correlated significantly with the resistance of *S. pneumoniae* as well. A new finding of this study is that RxIDs correlated better with the resistance of *S. pneumoniae* than total consumption of antibiotics expressed in DID and significant correlations exist between use of broad-spectrum penicillins expressed in PID and RxIDs.

## 1. Introduction

*Streptococcus pneumoniae* (also known as pneumococcus) is an important pathogen that causes severe infections in children and older adults worldwide [1]. Since 2017, *S. pneumoniae* has been listed as one of the twelve priority pathogens by the World Health Organization (WHO) [2]. Rising rates of resistance to penicillin and other antibiotics, as well as the continuing high burden of disease, have renewed interest in antibiotic therapy and prevention [3,4,5,6]. Antimicrobial resistance (AMR) is a threat to public health systems around the world [7,8]. Antibiotic use is a known risk factor for the emergence of antibiotic resistance, but demonstrating a causal relationship between antibiotic use and resistance is challenging. Several studies have found associations between total antibiotic and penicillin use with *Streptococcus pneumoniae* resistance to penicillin [9,10,11,12,13]. In contrast a study published by Olesen et al. showed weak or no correlations between total antibiotic use and the use of β-lactams with resistance to *S. pneumoniae* [14]. In all these studies, the only metric of antibiotic consumption was daily doses per 1000 inhabitants per day (DID and “old” DDDs were used [10,13,14]. The Collaborating Centre for Drug Statistics Methodology of the World Health organization (WHO-CC-DSM) recently changed some DDD definitions. From 1st January 2019, DDD measurement was revised for several Anatomical Therapeutic Chemical (ATC) classification codes for drugs commonly used in ambulatory care (e.g.,oral drugs, amoxicillin (J01CA04), amoxicillin/clavulanic acid (J01CR02)] and midecamycin (J01FA03) [15]. As a consequence of these changes there is a need to see the effect of antibiotic consumption expressed in new DIDs on the development of resistance. There is also a lack of long-term studies that correlate different metrics of antibiotic consumption and resistance of invasive *S. pneumoniae*. DDD is standardized tool for drug utilization research allowing presentation, comparison and benchmarking of drug consumption statistics at international and other levels. The feasibility and usefulness of this standardized metric strongly favours its complementation with another metric.

The present study aims to investigate the correlation between total antibiotic, penicillin and separately broad-spectrum penicillins consumption expressed in three metrics; DID the WHO version of 2019 (new version) and 2018 (old version), number of prescriptions per 1000 inhabitants per year (RxIDs) and number of packages per 1000 inhabitant per day (PIDs) with the resistance of invasive *S. pneumoniae* in Slovenia in the period from 2000 to 2018. Understanding the links between antibiotic use and *S. pneumoniae* resistance is critical for improving guidelines for therapy and vaccination strategy in Slovenia.

## 2. Results

Slovenia is a small country with a population of roughly 2.08 million inhabitants, according to the 2019 census [16]. In Slovenia, virtually all residents (>99%) have mandatory basic health insurance, and a prescription is required for any type of antibiotics. In addition, physicians prescribe antibiotics only for humans. Data on the number of antibiotic packages, the cost of antibiotics, the age and sex of patients and the identity number of the physicians and health care institutions prescribing antibiotics have been collected and published annually since 1976. In the period of 2000–2018, data on outpatient antibiotic use were collected using the ATC classification (ATC/defined daily doses (DDD) classification (WHO version 2018 and 2019) [15,17] The number of prescriptions for insured individuals and out-of-pocket paid antibiotics (“white” prescriptions for uninsured persons and prescriptions before travel) were provided by the National Institute of Public Health of Slovenia (NIPH) and Health Insurance Institute of Slovenia (HIIS). National antimicrobial stewardship activities were recently published [18,19]. In Slovenia, the surveillance of invasive diseases in children caused by *S. pneumoniae* has been continuously monitored since 1993 and in adults since 1996 [12,13,20].

### 2.1. Consumption of Antibiotics and Resistance of Invasive S. pneumoniae to Penicillin

During the study period of nineteen years, the prevalence of penicillin resistance of invasive *S. pneumoniae* decreased by 47.1% from 19.1% (21/110) in 2000 to 10.1% (28/276) in 2018. (Figure 1A). In the same period, the total consumption of antibiotics expressed in DID (WHO version 2019) decreased by 32.8% from 17.36 DID to 11.66 DID and by 28.7% from 20.02 DID to 14.27 DID (WHO version 2018), respectively (Figure 1A). A 12.4% (from 7.84 DID to 6.87 DID) and 9.3% (from 10.43 DID to 9.46 DID) decline in the consumption of penicillins, expressed in DID (for both WHO versions of DDDs), was identified. The consumption of extended-spectrum penicillin (J01CA) and co-amoxiclav (J01CR02) expressed in DIDs (WHO version 2019) decreased by 0.2% (from 5.20 DID to 5.19 DID) or no change in DDDs (WHO version 2018) (7.78 DID) (Figure 1B). Higher decrease of consumption expressed in RxIDs was observed for total antibiotics (J01) by 33.7% (from 724.5 to 480.6 RxIDs), penicillins (J01C) by 21.6% (from 364.7 to 285.8 RxIDs) and extended-spectrum penicillins and co-amoxiclav (J01 CA and J01CR 02) by 15.1% (from 269.11 to 228.4) (Figure 2).

The community consumption of antibiotics for systemic use (J01), β-lactams, penicillins (J01C) and extended-spectrum penicillins (J01CA) together with penicillins with β-lactamase inhibitors (CR02), expressed in the number of prescriptions/1000 population/year (RxIDs) and penicillin resistance of *Streptococcus pneumoniae* is shown in Figure 2.

The total consumption of antibiotics (J01) expressed in PID decreased by 36.5% (from 2.88 to 1.83 PID), for penicillins (J01C) by 23.0% (from 1.39 to 1.07 PID and extended-spectrum penicillins and co-amoxiclav (J01 CA and J01CR 02) by 22.2% (from 1.08 to 0.84 PID). (Figure 3). Number of DDDs of J01 per package increased by 5.8% (from 6.02 to 6.37 DDDs) and for penicillins (J01C) by 15% (from 5.6 DDDs to 6.4 DDDs), respectively. In contrast the number of DDDs of broad-spectrum penicillins (J01CA and J01CR02) decreased by 7.0% (from 5.88 DDDs to 5.29 DDDs) respectively.

### 2.2. Correlation between Consumption of Total Antibiotics for Systemic Use (J01), Penicillins (J01C) and Broad-Spectrum Penicillins (J01 CA + J01 CR02) and Resistance of Invasive S. pneumoniae to Penicillin (Intermediate and Resistant)

Correlations between the total use of antibiotics for systemic use (J01), penicillins (J01C), extended-spectrum penicillin (J01CA) and co-amoxiclav (J01CR02) expressed in DDDs WHO version 2018 and 2019 in RxIDsand in PID and resistance of invasive strains of *S. pneumoniae* in Slovenia 2000–2018 are summarised in Table 1.

### 2.3. Resistance of Invasive S. pneumoniae to Penicillin and Co-resistance in Penicillin-resistant S. pneumoniae Isolates

As seen in Figure 4A, sixty different serotypes were identified in the 4220 isolates. The most common one was serotype 3 (*n* = 623; 14.8%), followed by serotype 14 (*n* = 593; 14.1%), serotype 1 (*n* = 407; 9.6%),9V (*n* = 278; 6.6%), 4 (*n* = 270; 6.4%), 7F (*n* = 239; 5.7%), 19A (*n* = 194; 4.6%), 23F (*n* = 193; 4.6%), 19F (*n* = 150; 3.6%), 6B (*n* = 146; 3.5%), 6A (*n* = 143; 3.4%), 18C (*n* = 107; 2.5%), 22F (*n* = 100; 2.4%).

Among the 560 (13.3%) isolates less susceptible to penicillin, 29 different serotypes were determined (Figure 4A). Serotype 14 was the most prevalent (*n* = 176; 31.4%), followed by 19A (*n* = 117; 20.9%), 9V (*n* = 60; 10.7%), 19F (*n* = 53; 9.5%), 6B (*n* = 38; 6.8%), and 23F (*n* = 31; 5.5%). The most common co-resistance to oral antibiotics used in outpatients was resistance to co-trimoxazole (*n* = 369; 65.8%), erythromycin (*n* = 227; 40.5%) and tetracycline (*n* = 199; 35.5%). Among the combinations of co-resistance to penicillin, penicillin plus co-trimoxazole was the most prevalent (*n* = 87; 15.5%), followed by penicillin plus cefuroxime with co-trimoxazole (*n* = 52; 9.3%) (Figure 4B).

## 3. Discussion

The study shows a very high correlation between total use of antibiotics for systemic use (J01) expressed in RxIDs (R = 0.86), PID (R = 0.85) and DIDs version 2019 (R = 0.80) and resistance of invasive *S. pneumoniae* isolates. Total consumption expressed in DDDs version 2018 showed lower correlation (R = 0.78). The results are in agreement with previous studies that reported a significant correlation between total antibiotic use, expressed as DID, and resistance of *S. pneumoniae* [12,13].

A new finding of our study is that the consumption expressed in RxIDs had a higher correlation with the resistance of *S. pneumoniae* than the consumption of total use of antibiotics expressed in both DIDs. A very high correlation was also found between the use of penicillins, expressed as PID (R = 0.84) and RxIDs (R = 0.82) and lower (R = 0.59) correlation with DDDs version 2019. Nonsignificant correlation was found with DDDs version 2018 (R = 0.45, *p* = 0.05). In the current study, a lower correlation was found (R = 0.59 and R = 0.45 respectively) between use of penicillins expressed in DIDs and resistance of *S.pneumoniae*. Goossens et al. reported a Spearman’s coefficient of 0.84 and van de Sande Bruinsma a coefficient of 0.78 [12,13]. Our data are in agreement with the Olesen et al. study [14]. The highest correlation (R = 0.84) between use of β-lactams expressed in PID and resistance of *S. pneumoniae* is in accordance with the Bruyndonckx study [21] If we have no substantial changes in the amount of DDDs per package, PID is a good metric for antibiotic consumption in outpatients [22]. DID and PID, are used for a better understanding and interpretation of outpatient antibiotic use and its relation to resistance. RxIDs counts number of treatments independently of the prescribed doses and is used in many countries. It is useful in countries where OTC antibiotic use is not available [23]. A significant correlation (R = 0.72), (*p* < 0.001) and (R = 0.57), (*p* < 0.05) between consumption of extended spectrum penicillin (J01CA) and penicillins with β-lactamase inhibitors (CR02), expressed in PID and RxIDs and resistance of *S.pneumoniae* was found as well. Previous studies analysed resistance of *S. pneumoniae* among different countries, however this study analyses the long-term correlation between the use of antibiotics and resistance of *S. pneumoniae* in one country [12,13]. The selective pressure of antibiotics is the most important driver of antimicrobial resistance, although the consumption of antibiotics does not always correlate with the prevalence of resistance. EARS-Net data from 2018 and antibiotic consumption in ESAC-Net data from 2017 show that there are always exceptions when comparing antibiotic consumption and resistance [24,25]. Antibiotic selective pressure, clonality of resistant clones, co-selection and the fitness cost of resistance are important factors [23,25]. The spread of resistant clones is found in many studies [26,27,28,29,30]. In this study, a large number of different serotypes were found among penicillin-resistant isolates, suggesting the absence of clonal spread. In Slovenia, there is a decrease in the selective pressure of penicillin (12.38%) and co-selection with decreased use of other classes of antibiotics. The consumption of co-selective antibiotics, used in outpatients, co-trimoxazole (45.4%), macrolides (55.9%) and tetracyclines (77.6%) substantially decreased.

### Limitations of the Study

In Slovenia the universal vaccination in the national vaccine recommendations programme was introduced in 2015 and might have an influence on the prevalence of resistance of *S*. *pneumoniae*. Moderate coverage of vaccination (49–55%), low percentage of isolates (22%) in children and nonsignificant influence on the incidence of *S. pneumoniae* invasive infections diminish the influence of vaccination on the prevalence and resistance of *S. pneumoniae* [31].

## 4. Materials and methods

### 4.1. Collection of Isolates and Serotyping

Isolates of *S. pneumoniae* obtained from sterile body sites from patients with suspected invasive infection were collected from all Slovenian microbiological laboratories in the Department for Public Health Microbiology, National Laboratory of Health, Environment and Food in Ljubljana. The isolates were identified by classical colony morphology and hemolysis on blood agar and further tested for optochin susceptibility (Optochin Disc, Oxford, UK) and bile solubility. A total of 4223 isolates of *S. pneumoniae* were collected from 2000 to 2018. Of these, 930 (22%) were obtained from children (0–14 years) and 3293 (78%) from adults. Resistance was determined in 4208 isolates, as 15 isolates died off. The number of isolates per year varied from 93, in 2002, to 330, in 2015. Isolates were serotyped by Neufeld Quellung reaction with antisera (Statens Serum Institut, Copenhagen, Denmark).

### 4.2. Antibiotic Susceptibility Testing

Antibiotic susceptibility testing was performed using the disk diffusion method for oxacillin (screening test), erythromycin, clindamycin, vancomycin, and rifampicin. Etest (bioMérieux, Marcy-l’Étoile, France) was performed to determine the minimum inhibitory concentrations (MIC) of penicillin, cefotaxime, ceftriaxone, cefuroxime, erythromycin, tetracycline, chloramphenicol, and trimethoprim/sulfamethoxazole. For isolates up to 2013, the recommendations of Clinical Laboratory Standards Institute (CLSI) were followed; for isolates from 2014 onwards, the recommendations of European Committee at Antimicrobial Susceptibility Testing (EUCAST) were followed. *S. pneumoniae* ATCC 49619 isolate was used as quality control [32,33]. Penicillin resistant strains refer to *S. pneumoniae* isolates that have MIC values to benzylpenicillin above 0.06 mg/L. MIC data for co-resistance were analyzed according to EUCAST 2018 (v 8.1) using the following breakpoints: benzylpenicillin S ≤ 0.06 > 0.06 R, cefuroxime S ≤ 0.5 > 1R, cefotaxime S ≤ 0.5 > 2 R, ceftriaxone S ≤ 0.5 > 2R, erythromycin S ≤ 0.25 > 0.5 R, tetracycline S ≤ 1 > 2 R, chloramphenicol S ≤ 8 > 8R, trimethoprim-sulfamethoxazole S ≤ 1 > 2R.

### 4.3. Data Analysis

Statistical analyses were performed using the free software R (R Core Team 2019, Vienna, Austria) [34]. Spearman’s rho2 rank correlation coefficients at a 95% confidence level were calculated to infer possible monotonic associations between *S.pneumoniae* resistance and total antibiotic and penicillin consumption expressed with DDDs (WHO versions of 2018 and 2019), RxIDs and PIDs.

## 5. Conclusions

Data from the national long-term survey shows very high correlation between total antibiotic use expressed in RxIDs, PIDs and DIDs, WHO version 2019, and high correlation with DIDs WHO version 2018 with resistance of *S.pneumoniae* to penicillin. A new finding of this study is that the consumption expressed in RxIDs has a higher correlation with the resistance of *S. pneumoniae* than consumption of total use of antibiotics expressed in both DIDs. A significant correlation was also found between the use of penicillins expressed in PID and RxIDs and use of broad spectrum penicillins expressed in PID and RxIDs and resistance of *S.pneumoniae* respectively. Based on the stated results, RxIDs should be considered to be added to DID to monitor outpatient’s antibiotic consumption. Recently, the number of prescriptions per defined population was included in consensually validated metrics to assess the quantity of antibiotic use in the outpatient setting, enabling (inter)national comparisons [23]. To reduce the prevalence of resistance of *S. pneumoniae* to penicillin reduced number of prescriptions of all antibiotics and reduction of penicillins and broad-spectrum penicillins use is needed.

## Figures and Tables

**Figure 1 antibiotics-10-00758-f001:**
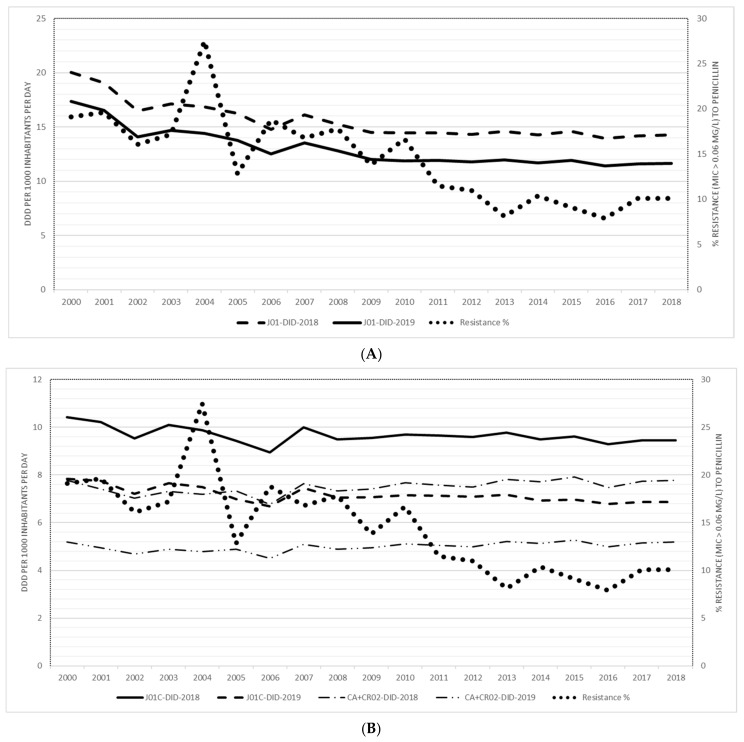
(**A**) The community consumption of the total of antibiotics for systemic use (J01), expressed in DDDs WHO version 2019 and 2018 and penicillin resistance of invasive *Streptococcus pneumoniae*. (**B**) The community consumption of β-lactams, penicillins (J01C) and extended-spectrum penicillins (J01CA) and penicillins with β-lactamase inhibitors (CR02), expressed in DDDs WHO version 2019 and 2018 and penicillin resistance of invasive *Streptococcus pneumoniae*.

**Figure 2 antibiotics-10-00758-f002:**
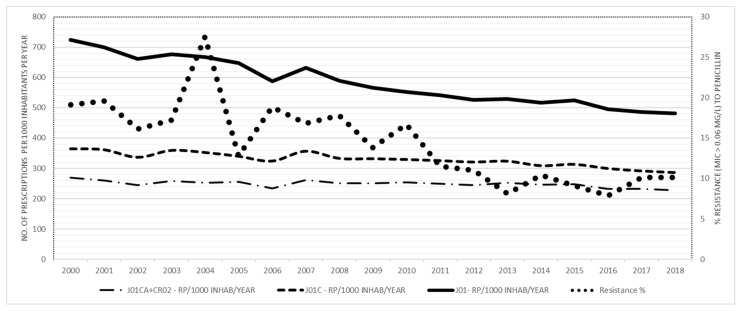
The community consumption of antibiotics for systemic use (J01), β-lactams, penicillins (J01C) and extended-spectrum penicillins (J01CA) + penicillins with β-lactamase inhibitors (CR02), expressed in the number of prescriptions/1000 population/year (RxIDs) and penicillin resistance of invasive *Streptococcus pneumoniae* are shown.

**Figure 3 antibiotics-10-00758-f003:**
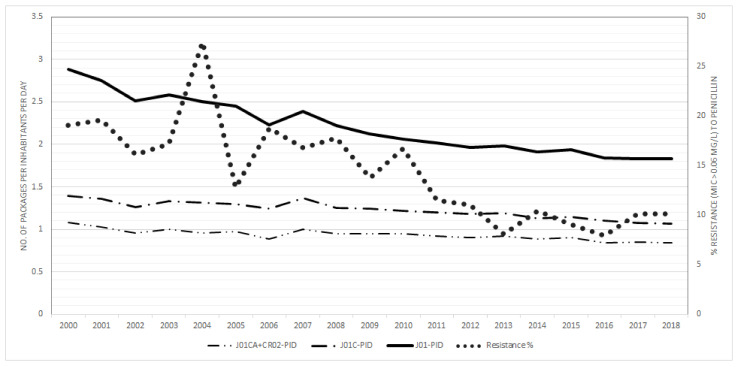
The community consumption of antibiotics for systemic use (J01), β-lactams, penicillins (J01C) and extended-spectrum penicillins (J01CA) + penicillins with β-lactamase inhibitors (CR02), expressed in the number of packages/1000 population/day (PID) and penicillin resistance of invasive *Streptococcus pneumoniae* are shown.

**Figure 4 antibiotics-10-00758-f004:**
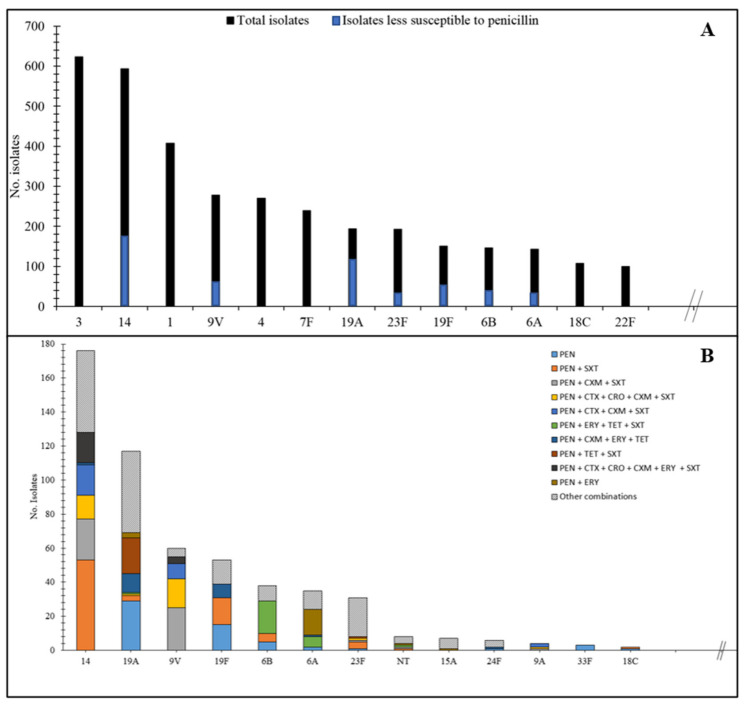
(**A**) Distribution of the most frequent serotypes among all 4220 (black bars) and the 560 (blue bars) less susceptible invasive *S. pneumoniae* isolates (MIC values to benzylpenicillin above 0.06 mg/L) and (**B**) Most frequent penicillin co-resistance combinations found in the most frequent 13 serotypes from the 560 less susceptible invasive *S. pneumoniae* isolates. * PEN-Pencillin, SXT-Trimethoprim-sulfamethoxazole, CXM-Cefuroxime, CTX-Cefotaxime, CRO-Ceftriaxone, ERY-Erythromycin, TET-Tetracycline.

**Table 1 antibiotics-10-00758-t001:** Spearman’s rho2 rank correlation coefficients between total antibiotic use of antibiotics for systemic use (J01), use of penicillins (J01C) use of broad-spectrum penicillins (J01CA and J01CR02) expressed in defined daily doses (DDD) WHO version 2018 and 2019, RxIDs and PIDs and resistance of invasive strains of *S. pneumoniae* in Slovenia.

Consumption in the Population	Spearman’s rho2 Rank Correlation Coefficient			
	DID WHO 2018	DID WHO 2019	RxIDs	PIDs
Total (J01) consumption vs resistance in total population	0.78 ***	0.80 ***	0.86 ***	0.85 ***
Consumption of beta-lactams, penicillins (J01C)	-	0.59 **	0.82 ***	0.84 ***
Consumption of extended spectrum penicillin (J01CA) and penicillin with β-lactamase inhibitors (CR02)	−0.58 **	−0.57 *	0.57 *	0.72 ***

Correlation is significant at the * 0.05 level (2-tailed), ** 0.01 level (2-tailed) and *** 0.001 level (2-tailed). Only significant correlations are shown.

## Data Availability

(https://www.ecdc.europa.eu/en, accessed on 22 June 2021) consumption data ESAC-Net, antibiotic resistance data EARS-Net.

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
