# Peer review of "Correlation between Antibiotic Consumption and Resistance of Invasive Streptococcus pneumoniae"

_antibiotics, 2021, doi:10.3390/antibiotics10070758_

Round 1
Reviewer 1 Report
In this study, the authors tried to find a relationship between antibiotic consumption and antimicrobial resistance of S. pneumoniae.
Introduction
Introduction was mostly well written, covered the main background points and led up to the aim of the study. However, it is unclear why authors only analysed the total antibiotic usage and use of penicillins, and not the use of other groups of antibiotics used in S. pneumoniae treatment.
Methods
Authors presented the key element of the study design and adequately described the used methods. However, only Spearman's rho coefficient was used to determine the correlation. Perhaps authors could have also used linear regression analysis to supplement the previously mentioned statistical analysis.
Results and discussion
There is an issue with the figures 1 and 2. It is difficult to interpret the legends as the colours of the lines are so similar. Authors tried do differentiate the lines by using the different line types but the figure legend does not show those types so it does not help. Also, figures show consumption of antibiotics expressed as DIDa and PxIDa but the consumption expressed as PIDs was not shown.
Authors should try to explain why some metrics had better correlation than others. Could they be considered more adequate as a measure of consumption? How does that relate to the results of the previous studies that primarily used DIDs as a measure of consumption?
Are there any other factors that might have influenced the correlation, besides the vaccination? Was there any national antimicrobial stewardship programme implemented in that time, campaign to raise awareness towards this issue, change in legislation regarding drug prescribing practices?
Overall, this is not a novel study, but the topic is still relevant and it introduces some previously under-used metrics. It could be interesting to further analyse the adequacy of the new metrics in comparison to established ones. Also, this study should be relevant for the practitioners and policy makers in Slovenia.
Author Response
Reviewer 1
Response 1: Please provide your response for Point 1. (in red)
Thank you for useful suggestions for improvement and comments. Please find below my corrections :
-Introduction: we analysed the total antibiotic usage and use of penicillins.We have choosen the same groups of antibiotics as authors in previous studies (Ref.10,12,13). Total use of antibiotics and use of penicillins were associated with Streptococcus pneumoniae resistance to penicillin. High usage of macrolides is associated with resistance of Streptococcus pneumoniae to macrolides.Hih usage of quinolones with the resistance to quinolones. Use of co-selective antibiotics (co-trimoxazol, macrolides and tetracyclines is shown i the paragraph Discussion line 170 ,171).
Methods: As in previous studies( ref 10,12,13,14) we used the same statistical method.
Results: Figures 1 and 2 are corrected. In Figure 1 we included the consumption od total antibiotics, penicillins and broad-spectrum penicillins expressed in PID.
In the paragraph Introduction we added the sentence on. DDD is standardized tool for drug utilization research allowing presentation, comparison and benchmarking of drug consumption statistics at international and other levels. The feasibility and usefulness of this standardized metric strongly favours its complementation with another metric. Most authors emphasize the need to combine different quantity metrics to optimize interpretation of the volumes of antibiotic use as each of the single metrics had some pitfalls in interpretation. In Discussion we added DID and PID, should be used for a better understanding and interpretation of outpatient antibiotic use and its relation to resistance.RxIDs counts number of treatments independently of the prescribed doses and is used in many countries. It is a good proxy for packages.
National antimicrobial stewardship programme was recently presented in two studies (added references 18,19)
19.Cizman, M.; Plankar, Srovin, T.; Beovic, B.; Vrdelja, M.; Bajec, T.;Blagus, R. European Antibiotic Awareness Day (EAAD): any impact on antibiotic consumption and public awareness in Slovenia? J Antimicrob Chemother. 2018,73, 2567-2572. doi: 10.1093/jac/dky206.
18.Fürst, J.; Čižman, M.; Mrak, J.; Kos, D.; Campbell, S.; Coenen, S.; Gustafsson, L.L.; Fürst, J.; Godman, B. The influence of a sustained multifaceted approach to improve antibiotic prescribing in Slovenia during the past decade: findings and implications. Rev Anti Infect Ther. 2015, 13, 279-89. doi: 10.1586/14787210.2015.990381.
Reviewer 2 Report
The article analyzes the correlation between total antibiotic and penicillins consumption expressed in three different metrics with the resistance of invasive S. pneumoniae in Slovenia in the period from 2000 to 2018. The field of research is certainly important and relevant, and the correlations found deserve the attention of the scientific community.
However, the article is written in such a concise form that it remains unclear on the basis of what data the conclusions of this study are made. The section "Results" consists of two parts. The first of them describes the consumption of antibiotics and the resistance of invasive S. pneumoniae to penicillin according to WHO, the second describes correlations between consumption of different antibiotics and resistance of invasive S. pneumoniae to penicillin. It is not clear from what data regularities described and illustrated (figures 3 and 4) in the second part (2.2) are derived. If this is the experimental data of the Authors themselves, then they deserve a more detailed description with the tables of MIC data (possibly in Supplementary materials) and other experimental details. If this data have been obtained earlier, the sources of this data should be clearly cited. In the latter case, if the experimental data do not relate to this study, testing of antibiotic sensitivity (4.3) should not be given in the section "Materials and methods". These tests could be described briefly with the necessary references in the Results. Also, most of the text that is currently present in the Materials and Methods section should be moved to the Results section: 4.1 (in full), 4.2 (beginning), 4.3 (see above).
Some more specific comments.
- Figures 1 and 2 can be combined and presented as separate panels of the same figure. The same applies to Figures 3 and 4.
- It is necessary to give clear criteria for what is "less susceptible" isolates.
- Figure 4 contains many abbreviations of antibiotics names, Authors should decipher the abbreviations in the figure legend.
- The section Authors contributions contains an indication of the contribution of three Authors of the article, and nothing is written about the other three.
- The list of References needs to be edited.
The article leaves an ambiguous impression and can be published only after a significant revision.
Author Response
Reviewer 2
Response 1: Please provide your response for Point 1. (in red)
Thank you very much for your useful suggestions and comments.
Please find below my corrections.
-Slovenia monitor the incidence, resistance, serotypes of invasive strains of S.pneumoniae in children since 1993 and additionally from adults since 1996. (Ref.18). Monitoring is a special interest of the NIPH to show the national problem of pneumococcal infections to introduce vaccination.Slovenia is also participant of the EARS and EARS-Net since 2000 and in ESAC and ESAC- Net since the beginning in.2001..We are using their methodology what is described in REf 12 and 13. Source of isolates are usually hospitalized patients with clinical picture of suspected invasive S.pneumoniae infection. CLSI and since 2014 EUCAST standards were used for antibiotic susceptibility testing.Less susceptible strains to penicillin (MIC > 0.06mg/L) were described in 4.3 line 219 and 220..
All figures are changed to be more clear.
In the Figure4 the criteria what is less susceptible is added
Figure 4 abbreviations are clarifyed
Author contributions is improved
References are edited
Round 2
Reviewer 1 Report
As the authors have accepted all suggestions and made corrections, I have no further objections to the text of the manuscript. The manuscript has been much improved and it is now simple and easy to read.
Author Response
The Reviewer 1 accepted our improvements and has not further objections.
Reviewer 2 Report
The Authors took into account some comments and made appropriate corrections in the manuscript. However, the Authors did not take into account the comments regarding the presentation of the Results and Methods. In the structure of the article, the Results section precedes the Methods section. At the same time, the information contained in the Methods (4.1, beginning of the 4.2) would look better in the Results section. At the moment, in the course of reading the text, it is completely unclear what data the authors discuss in the Results chapter. Also, it is not necessary to give in the chapter Methods methods that the Authors themselves did not use in this study (4.3), this information can also be reported in the chapter Results. Since the Authors have not commented on these comments, it seems that the Authors do not agree with them.
I believe that the article should be finalized before its publication.
Author Response
Reviewer 2
I do appreciate the comments of the reviewer 2
Point 1.In the Instructions for Authors. Manuscript preparation the Reseach manuscript sections for Antibiotics journal should be : Introduction,Results, Discussion, Material and Methods and conclusion (optional), I followed these instructions.
Point 2.The information contained in the Mehods (4.1) was trasfered in the chapter Results .The beginning of the 4.2 was also moved to chapter Results and I added 2 Ref.(12.13 )
Point 3.I am very sad that the the Results are not clear.We wanted to investigate the correlation between the national outpatients consumption of antibiotics including usage of some classes and resistance of S.pneumoniae.The authors of the manuscript collected the national data of resistance of S.pneumoniae and national data of antibiotic usage in the community for very long period.We included three metrics (DID,PID,RxIDs) of antibiotic consumption (it is better three metrics than only one) including old and new DDDs. Such data are very useful for policy maker to decrease the resistance of the pneumococcus.
Point 4 In 4.4 we describe the methodology and MIC of antibiotic susceptibility.The results of antibiotic susceptibility are in the chapter Results
I hope that I replied to your comments
Kind regards
Milan Cizman
Round 3
Reviewer 2 Report
The Authors made necessary corrections to the manuscript. The manuscript can be accepted in its present form after correcting typos (in particular, in lines 214 and 227)